# Mechanistic Studies of Improving Pt Catalyst Stability at High Potential via Designing Hydrophobic Micro-Environment with Ionic Liquid in PEMFC

Lei Huang [1,2], Fen Zhou [1,3], Hui Zhang [1,2], Jinting Tan [1,2,3,*] and Mu Pan [1,2,3,*]

1 State Key Laboratory of Advanced Technology for Materials Synthesis and Processing, Wuhan University of Technology, Wuhan 430070, China
2 Hubei Key Laboratory of Fuel Cell, Wuhan 430070, China
3 Foshan Xianhu Laboratory of the Advanced Energy Science and Technology Guangdong Laboratory, Xianhu Hydrogen Valley, Foshan 528200, China
* Correspondence: tanjinting@whut.edu.cn (J.T.); panmu@whut.edu.cn (M.P.); Tel.: +86-180-0719-6313 (J.T.); +86-135-0711-6428 (M.P.)

**Abstract:** Recently, the focus of fuel cell technologies has shifted from light-duty automotive to heavy-duty vehicle applications, which require improving the stability of membrane electrode assemblies (MEAs) at high constant potential. The hydrophilicity of Pt makes it easy to combine with water molecules and then oxidize at high potential, resulting in poor durability of the catalyst. In this work, an ionic liquid [BMIM][NTF$_2$] was used to modify the Pt catalyst (Pt/C + IL) to create a hydrophobic, antioxidant micro-environment in the catalyst layer (CL). The effect of [BMIM][NTF$_2$] on the decay of the CL performance at high constant potential (0.85 V) for a long time was investigated. It was found that the performance attenuation of Pt/C + IL in the high-potential range (OCV 0.75 V) was less than that of commercial Pt/C after 10 h. The Pt-oxide coverage test showed that the hydrophobic micro-environment of the CL enhanced the stability by inhibiting Pt oxidation. In addition, the electrochemical recovery of Pt oxides showed that the content of recoverable oxides in Pt/C + IL was higher than that in commercial Pt/C. Overall, modifying the Pt catalyst with hydrophobic ionic liquid is an effective strategy to improve the catalyst stability and reduce the irreversible voltage loss caused by the oxide at high constant potential.

**Keywords:** ionic liquid; catalytic applications; oxygen coverage; catalyst stability; MEAs

## 1. Introduction

As a clean, efficient and green power source, proton-exchange membrane fuel cells (PEMFCs) are an indispensable part of the renewable-energy-driven future. Especially in heavy-duty transportation, heavy-duty application requires long-term and high-efficiency operation, and high efficiency means high working potential [1]. The U.S. Department of Energy (DOE) proposes that PEMFCs work at operating potentials over 0.9 V and a working time of more than 25,000 h during heavy-duty application [2,3]. Therefore, improving the durability at high potential is an important goal of PEMFC development at present. In addition, such a high working potential is just in the oxidation potential range of Pt, which is one of the main reasons for the performance degradation of the catalyst [4]. If the oxidation time is kept for just 60 s at a high potential of 1.1 V, the activity loss is 36.73% [5]. Additionally, studies have shown that once the platinum oxide is formed, 15–35% of the properties at different potentials cannot be recovered [6]. Accordingly, the instability of cathode Pt catalysts is one of the great challenges that limits the wide application of PEMFC technology.

According to Formulas (1) and (2) of Pt oxidation, the factors causing Pt oxidation can be divided into intrinsic factors and environmental factors.

$$Pt + H_2O \rightarrow PtOH + H^+ + e^- \qquad 0.8–1.08 \text{ V} \qquad (1)$$

$$PtOH \rightarrow OHPt \rightarrow PtO + H^+ + e^- \qquad > 0.85 \text{ V} \qquad (2)$$

$$PtO + H_2O \rightarrow PtO_2 + 2H^+ + e^- \qquad > 1.05 \text{ V} \qquad (3)$$

On the one hand, water is the major product of the PEMFC reaction, and the hydrophilicity of Pt makes it easy to bind with water molecules. Then, water becomes a major source of oxygenated species and inevitably leads to the oxidation of Pt [7–11]. On the other hand, when the voltage reaches above 0.8 V, the internal electronic structure of surface Pt atoms rearranges after the critical surface is covered by oxygen [12,13]. This contributes to the oxidation of Pt [4,14,15]. The oxidation behavior of Pt can seriously affect the utilization of Pt. Therefore, how to inhibit the formation of Pt oxides and how to restore the negative effects of Pt oxides have become hot topics [16–18].

According to the intrinsic causes of Pt oxidation, researchers have put forward many strategies to depress Pt oxidation. It was found that the usage of a Pt NPs (nanoparticles) catalyst with larger size is an effective way [19,20]. However, the specific surface area of catalyst particles is greatly reduced. Pt NPs designed with a specific crystal phase [21,22] change the electronic state of Pt and then improve the oxidation resistance and oxidation–reduction reaction (ORR) activity of Pt. Nevertheless, the residue of synthetic reactants greatly affects the reaction [23,24]. Another method to restrain the formation of oxide on the catalyst surface is alloying treatment of Pt [25,26], but leaching and dissolution of non-precious metal components [27] and instability of surface structure [28] seriously affect the service life of the catalyst.

Regarding the environmental factors relating to forming Pt oxide, the key is to discharge the product water from the surface of Pt NPs in time without sacrificing ORR activity [29]. It is necessary to create a hydrophobic micro-environment near the surface of the Pt particles. Some researchers have used planar macrocyclic compound t-BuTAP for hydrophobic treatment. However, the large size and poor stability prevent its application [30]. Furthermore, the catalyst layer (CL) has been doped with polytetrafluoroethylene (PTFE) [31–33] and fluorinated ethylene propylene (FEP) [34] hydrophobic polymers to improve the hydrophobicity of the CL. In earlier studies, PTFE was often used to modify CLs for hydrophobic treatment of Pt NPs. However, due to its lack of proton conductivity, this method was gradually phased out [32]. Ionic liquid (IL), as a small-molecule compound, generally contains hydrophobic anions (e.g., $PF^{6-}$ and $N(SO_2CF_3)^{2-}$) that make ILs appear hydrophobic [35,36]. At the same time, it can be directly adsorbed on the surface of Pt NPs in a ligand-like manner to fabricate a local hydrophobic environment on the Pt surface [37]. The properties of ILs are quite different from those of traditional aqueous and non-aqueous electrolytes. ILs contain pure cations and anions, which leads to unique IL–electrode interfaces that are more viscous than water so that the diffusion coefficient of redox species in the ILs differs from that of aqueous electrolytes [38]. The IL phase is supposed to partially replace water as the reaction media, and, more importantly, the created hydrophobic micro-environment at catalyst surfaces helps to preserve active sites by repelling water molecules from the product and aqueous electrolyte [39–41]. The promotional effect of ILs as electrolytes originates from the weak moisture-absorbing properties, high solubility of reactant gases and good mass transport properties, which are desirable for improving the kinetics of the electrode reactions. Due to their good conductivity and suitably viscosity, ILs have also been employed as a new kind of binder [42,43]. In 2007, a concept called 'solid catalysts with ionic liquid layers' (SCILL) was developed [35]. Since then, IL has gradually been used in PEMFCs, mainly to improve ORR performance [44–49]. Although the mechanism of IL improving ORR performance is still controversial, it is mainly believed to increase the oxygen permeability of the Pt surface [44–46]. A traditional Pt/C catalyst was modified with hydrophobic IL ([MTBD][NTF_2]), and it presented

an obvious improvement of ORR performance in the rotating disk electrode (RDE) [47]. Huang et al. [49] explored different hydrophobic ILs on Pt/C catalysts and determined that ILs can improve the cycle stability of Pt catalysts. After 5000 cycles in the range of 0.6–1.0 V, the CL with IL could effectively alleviate the current decay. However, these studies are not comparable to the high constant potential condition of PEMFCs in heavy-duty application, nor did they explore the influence of IL on the durability of the Pt/C catalyst in single cells. Moreover, the mechanism of improving the catalyst's durability via small-molecular, hydrophobic IL is still unclear.

This work aims to explore the influence mechanism of the hydrophobic micro-environment created by IL on Pt oxidation and the relation between Pt oxidation and durability at high constant potential. [BMIM][NTF$_2$] is chosen to modify the surface of the Pt catalyst. The influence of the IL on the performance degradation of the PEMFC catalyst under long-term operation conditions at high constant potential (0.85 V) is explored. The inhibiting effect of IL on Pt oxidation and the mitigation of IL on performance decay, as well as the electrochemical surface active area (*ECSA*) loss, are analyzed. In addition, the reversible performance loss caused by Pt oxidation is also studied. X-ray photoelectron spectroscopy (XPS) is applied to make clear the change of Pt oxidation in Pt/C and Pt/C + IL before and after the CV recovery procedure.

## 2. Results and Discussion

The miscibility of an ionic liquid with water is mostly determined by the associated anion. In the presence of water, water-immiscible ILs display an obvious structural rearrangement at the gas−liquid interface [50]. In addition, the activity of water is higher in the hydrophobic ionic liquid than in the hydrophilic [51]. In this way, water is distributed to the gas−liquid interface of 1-butyl-3-methylimidazolium bis((trifluoromethyl)sulfonyl)imide ([BMIM][NTF2]), forming a water network structure rather than directly acting on the surface of the Pt NPs. It is through this mechanism that IL forms a hydrophobic micro-environment on the Pt surface.

Water management is very important for PEMFCs. Figure 1 shows the contact angle of the catalyst surface before and after hydrophobic treatment. A 5 μL drop was used in contact angle measurements. The maximum contact angle of Pt/C MEA was 128.6°, while for Pt/C + IL it was 143.5°. The higher contact angle after treatment means the surface of the catalyst was less susceptible to water wetting, which increased the hydrophobicity of the Pt/C CL. The results show that the introduction of [BMIM][NTF$_2$] really increased the hydrophobicity of the Pt/C CL.

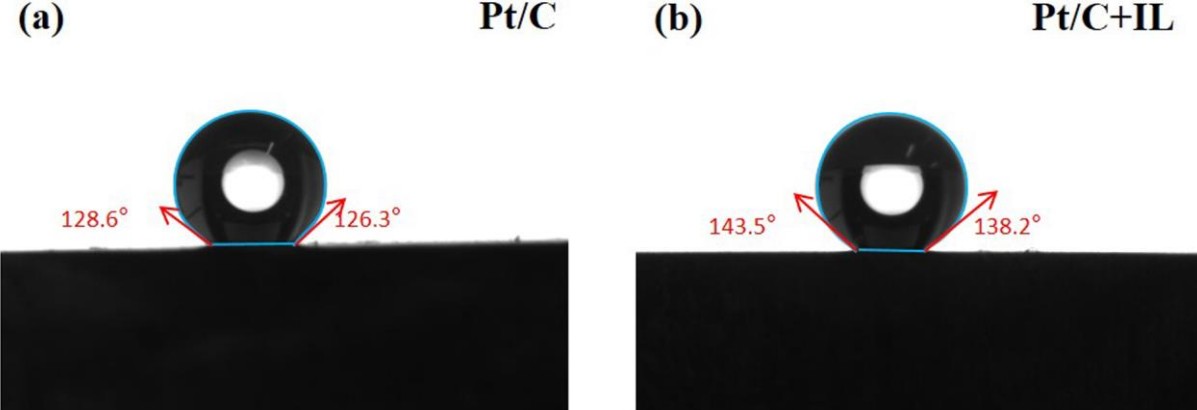

**Figure 1.** Contact angle of Pt/C (**a**) and Pt/C + IL (**b**).

To explore the influence of the hydrophobic micro-environment on the durability at 0.85 V, the Pt/C and Pt/C + IL MEAs were operated at the voltage continuously for 10 h, and the decay curves of current density with time were recorded, as shown in Figure 2. It

can be seen that the initial current density of the two MEAs did not defer too much, then dropped rapidly in the first 5000 s, then finally reached equilibrium after 10,000 s. The decay rate of Pt/C MEA was even up to 40% in the first 5000 s, and the current density declined from the initial 131.76 mA/cm$^2$ to 52.70 mA/cm$^2$ after 10 h, and the decay rate grew to 60%. However, the current density of Pt/C + IL MEA displayed a lower attenuation rate of 45%, and the current density at the end increased by nearly 20 mA/cm$^2$ compared with Pt/C MEA. Comparing the current attenuation trend before and after IL treatment, the formation of a local, hydrophobic micro-environment did improve the performance attenuation of the MEA.

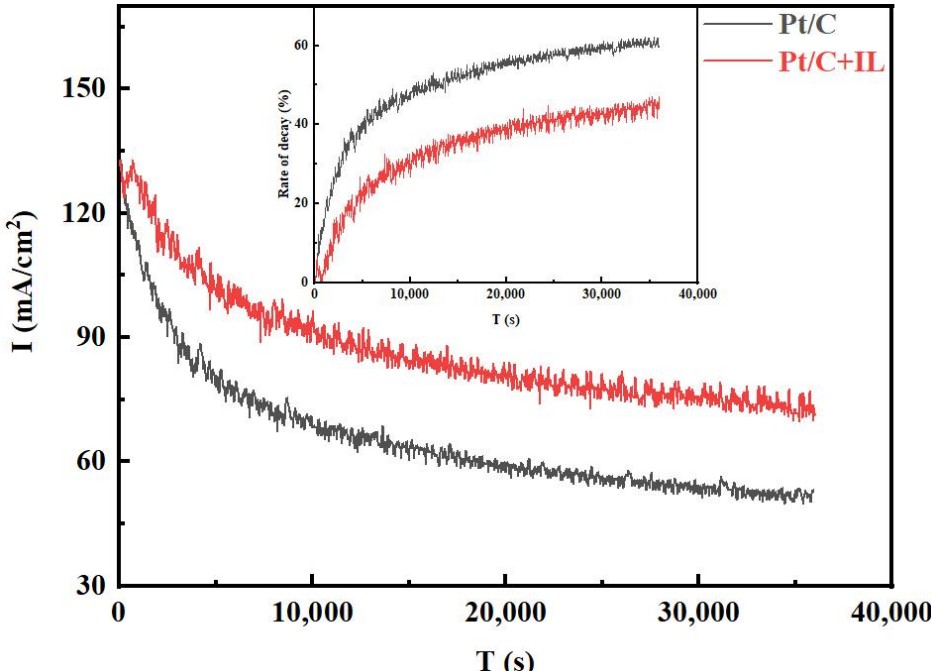

**Figure 2.** The I–t curves for Pt/C and Pt/C + IL (0.85 V, 10 h).

The hydrophobic micro-environment created by [BMIM][NTF$_2$] can help water molecules be removed quickly from the surface of Pt. It can reduce the inactive oxides produced by the binding of Pt and water molecules, protecting the active site from occupancy, which is an important factor for improving the stability of the catalyst. However, it only applies to the initial stage of oxidation [52]. When the potential is over 1.05 V or the oxidation occurs for a long time, the Pt surface forms more complex structures and more stable oxides [53], such as in Formulas (2) and (3).

The performance decay caused by Pt oxidation during fuel cell operation can be divided into recoverable performance loss (often denominated reversible degradation effects) and unrecoverable performance loss [54]. The irreversible performance loss can only be avoided by specific operation strategy and material improvement, while the reversible performance loss can be recovered by some procedures. In general, the oxides formed on platinum can be reduced to metallic platinum. The exact recovery conditions required depend on the type of platinum oxide formed [55,56]. Therefore, Pt oxides can be reduced in the following two ways depending on the surface coverage. When most of the surface is covered ($\theta > 60$), the electrochemical surface active area is covered, and a chemical reaction with hydrogen gas is required [57], as in Formula (4).

$$2PtO + H_2 \rightarrow 2PtOH \rightarrow PtO + Pt + H_2O \tag{4}$$

When the coverage rate is low ($\theta < 60$) and the surface contains reactive free Pt NPs, a hydrogen oxidation reaction (HOR) is used as the main reduction mode, according to Formula (5) [57].

$$PtO + 2H^+ + 2e^- \rightarrow Pt + 2H_2O \tag{5}$$

The combination of these two reactions can completely remove the recoverable oxide and, thus, fully recover the performance loss. Different publications have demonstrated that full recovery of performance losses can be achieved by injecting nitrogen into the cathode by cyclic voltammetry (CV) and reducing the potential to 0–100 mV with a moderate sweep rate of 10–50 mV/s [6,57–59]. To study the effect of [BMIM][NTF$_2$] on the oxidation of Pt, the Pt oxide coverage before and after the durability test was characterized. Electrochemical recovery was performed by the CV method [60], and the Pt oxide coverage after recovery was also analyzed. In Figure 3a, the CV curves before and after durability and after recovery can be observed. The hydrogen characteristic peaks at low potential hardly showed obvious changes, but clear changes were observed in the Pt oxide region at high potential. The oxide coverage value extracted from Figure 3a is concluded in Figure 3b. It can be seen from Figure 3b that the hydrophobic Pt/C + IL reduced the initial oxide coverage by nearly half. After 10 h of durability tests, the oxide coverage in the Pt/C CL increased to 146%, even more than 100%. It can be attributed to long-term oxidation, resulting in the generation of oxides such as PtO$_2$, which contribute an additional charge to oxygen reduction and lead to partial false oxide coverage [5]. It has been reported that PtO$_2$ means that oxygen atoms have entered the electron inner layer of Pt particles which is highly stable and is the culprit for the irreversible degradation of catalyst performance [61]. Comparatively, the Pt oxide coverage was only 41.9% for Pt/C + IL after 10 h at 0.85 V and decreased to 33.1% after CV recovery. Although CV treatment restored the oxide coverage of the Pt/C CL, 73.1% of the Pt was still covered by inactive oxides, which was more than twice than that of Pt/C + IL. Therefore, the hydrophobic micro-environment created by IL can effectively inhibit the formation of Pt oxides. It makes the intermediate adsorbed on Pt(111) species unstable, destroys the proportional relation between OH$_{ad}$ and O$_{ad}$ and improves the oxidation resistance of Pt [62]. At the same time, the formed oxides can be recovered to a great extent, which may be caused by the reduction of the bonding strength of Pt–H through the ligand effect [63]. In addition, it also shows that once the irreversible stable oxide is formed, it seriously hinders the proton conduction and leads to the degradation of performance [64]. The Pt oxide coverage test showed that the addition of IL can increase the proportion of recoverable oxides, which is of great significance for reducing the performance loss of CLs.

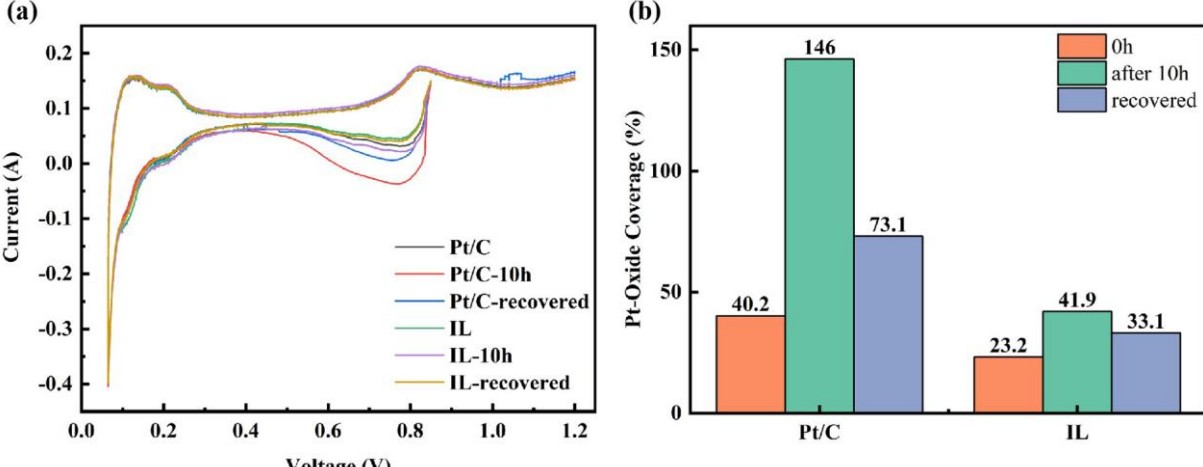

**Figure 3.** Pt oxide coverage curve of Pt/C and Pt/C + IL (**a**) and Pt oxide coverage statistics (**b**).

The polarization curves were recorded to further evaluate the effects of Pt oxides and the recovery procedure on cell performance, as exhibited in Figure 4a. The voltage at three characteristic current densities was selected for analysis, as shown in Figure 4b–d. The voltage of Pt/C decay from 0.831 V to 0.797 V at the current density of 200 mA/cm$^2$ had an attenuation rate which reached 4.09%, while the voltage attenuation rate of Pt/C + IL was only 1.89%. The voltage after CV recovery for Pt/C was 18 mV lower than that of Pt/C + IL at 200 mA/cm$^2$. In addition, the attenuation rates of Pt/C were 6.06% and 5.17%, respectively, at 1000 mA/cm$^2$ and 2000 mA/cm$^2$, while the attenuation rates of Pt/C + IL were 4.24% and 4.42%, respectively. The performance of Pt/C and Pt/C + IL presented a huge reduction after durability. After recovery, however, the irreversible loss decreased obviously with the increase in current density, and the voltage was completely recovered at 2000 mA/cm$^2$. Such a high working potential is just in the oxidation potential range of Pt; it can be summarized that the enhanced oxidation resistance of Pt by IL mainly works in the high-potential range. The effect of IL on performance attenuation in the low-potential region is not obvious. In addition, IL promotes the formation of recoverable Pt oxides, which means that adding an appropriate amount of IL to the CL can inhibit Pt from forming a stable oxide. At the same time, IL can also reduce the MEA performance loss caused by Pt oxidation to some extent.

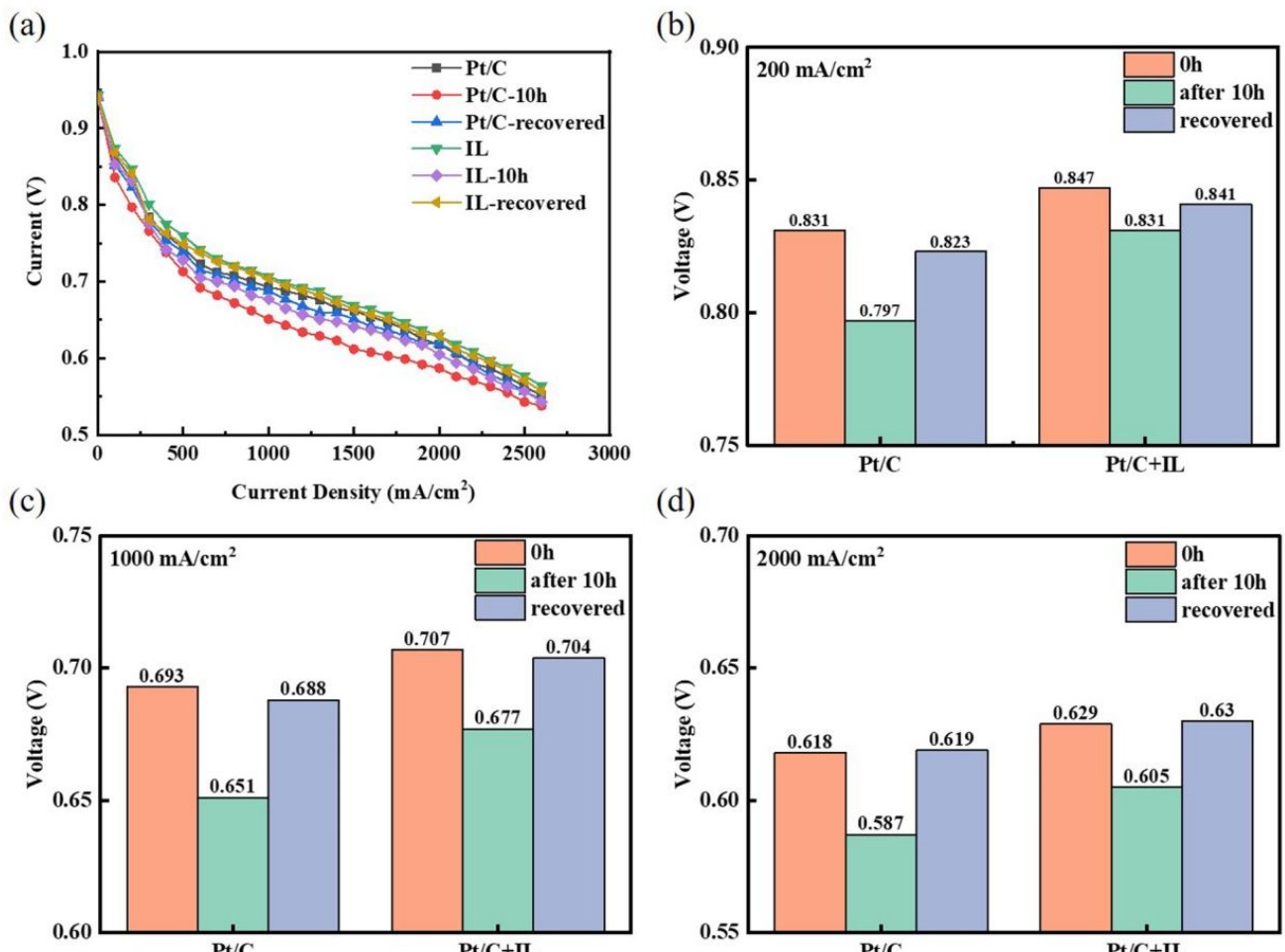

**Figure 4.** Polarization curves of Pt/C and Pt/C + IL (**a**); voltages at low (**b**), medium (**c**), and high (**d**) current densities.

The change of *ECSA* is an important indicator of catalyst corrosion. The CV curve in Figure 5 illustrates the inhibition effect of [BMIM][NTF$_2$] on the formation of Pt oxide from another point of view. Compared with the initial *ECSA*, Pt/C + IL was lower than Pt/C.

After the durability test, both the *ECSA*s slightly decreased, and Pt/C + IL almost reached the initial value after CV recovery. The *ECSA* of Pt/C + IL decreased from 54 $m^2/g_{Pt}$ to 51.7 $m^2/g_{Pt}$, reduced by 4.26% and was almost completely recovered, while the *ECSA* of Pt/C decreased from 60.2 $m^2/g_{Pt}$ to 49.8 $m^2/g_{Pt}$, reduced by 17.28% and still left 4% unrecoverable. On the one hand, IL itself may occupy the active site, resulting in a smaller initial *ECSA* [47]. On the other hand, IL inhibits the oxidation of Pt, which can prevent inactive oxidizing substances from occupying active sites. At the same time, the shedding of Pt oxide takes away part of the Pt [4]. Therefore, a hydrophobic micro-environment can also reduce the loss of Pt, making the *ECSA* decrease slowly. This is consistent with the test results in RDE presented in previous reference [47].

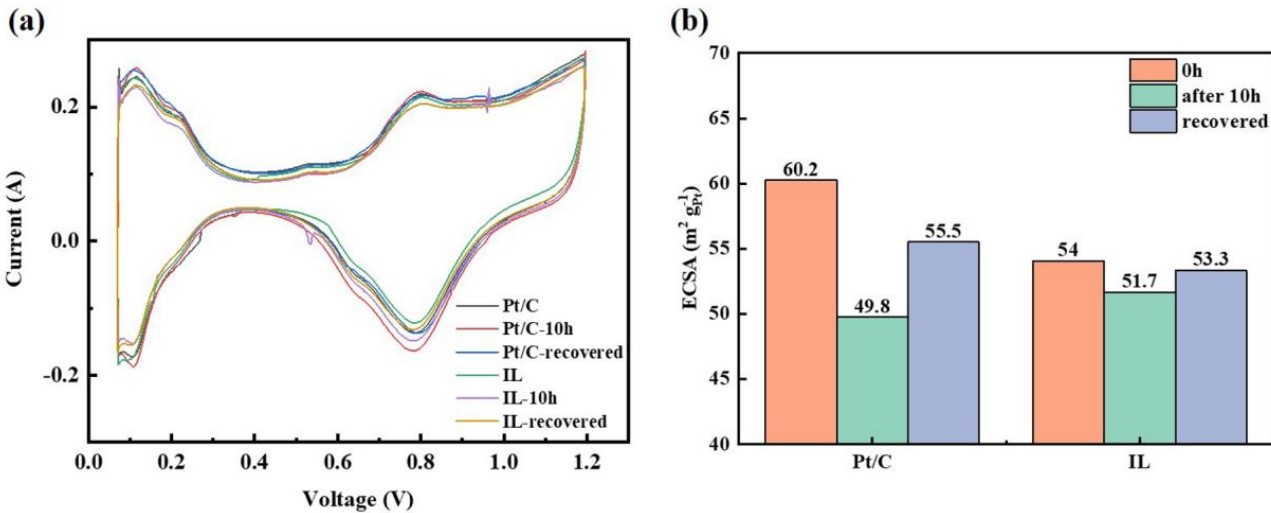

**Figure 5.** The CV curve of Pt/C and Pt/C + IL (**a**) and *ECSA* statistics (**b**).

In order to determine the Pt oxide in the CL more intuitively, the catalyst was analyzed by XPS. Figure 6a–f, respectively, show the curves of the Pt/C catalyst and Pt/C + IL catalyst before the durability test, after the durability test and after CV recovery. Pt (II) and Pt (IV) correspond to the different oxidation valence states of Pt. According to Formulas (1)–(3), the main divalent oxides of Pt are PtOH and PtO, and the tetravalent oxide is $PtO_2$. Without any experiment, the XPS maps of both Pt/C and Pt/C + IL contained Pt peaks, and their relative peak areas were similar. The total oxide content in Pt/C-0 h (38.42%) was higher than that in Pt/C + IL-0 h (35.69%). Table 1 shows the Pt oxide content of Pt/C and Pt/C + IL under three conditions. Among them, the relative peak areas of Pt (II) and Pt (IV) in Pt/C-10 h increased significantly and still could not be eliminated after CV recovery. After the 10 h durability test, the relative peak areas of Pt (II) and Pt (IV) reached 32.52% and 11.64%, respectively. Moreover, after the recovery procedure, the relative peak areas of Pt (IV) still remained at 9.39%. It could be considered that a stable and unrecoverable inactive oxidation substance was formed. Pt/C + IL confirmed the above discussion. The relative peak area of Pt (II) in Pt/C + IL changed little before and after the durability test. In addition, the peak of Pt (IV) completely disappeared after CV recovery. This explains why the performance of Pt/C + IL catalyst was good overall and how the performance was completely recovered after CV recovery. Compared with pure Pt, the overall oxidation state of Pt in a hydrophobic environment is low, and it is difficult to form stable oxide. It improves the stability of the catalyst and provides a guarantee for the long-term efficient operation of MEAs.

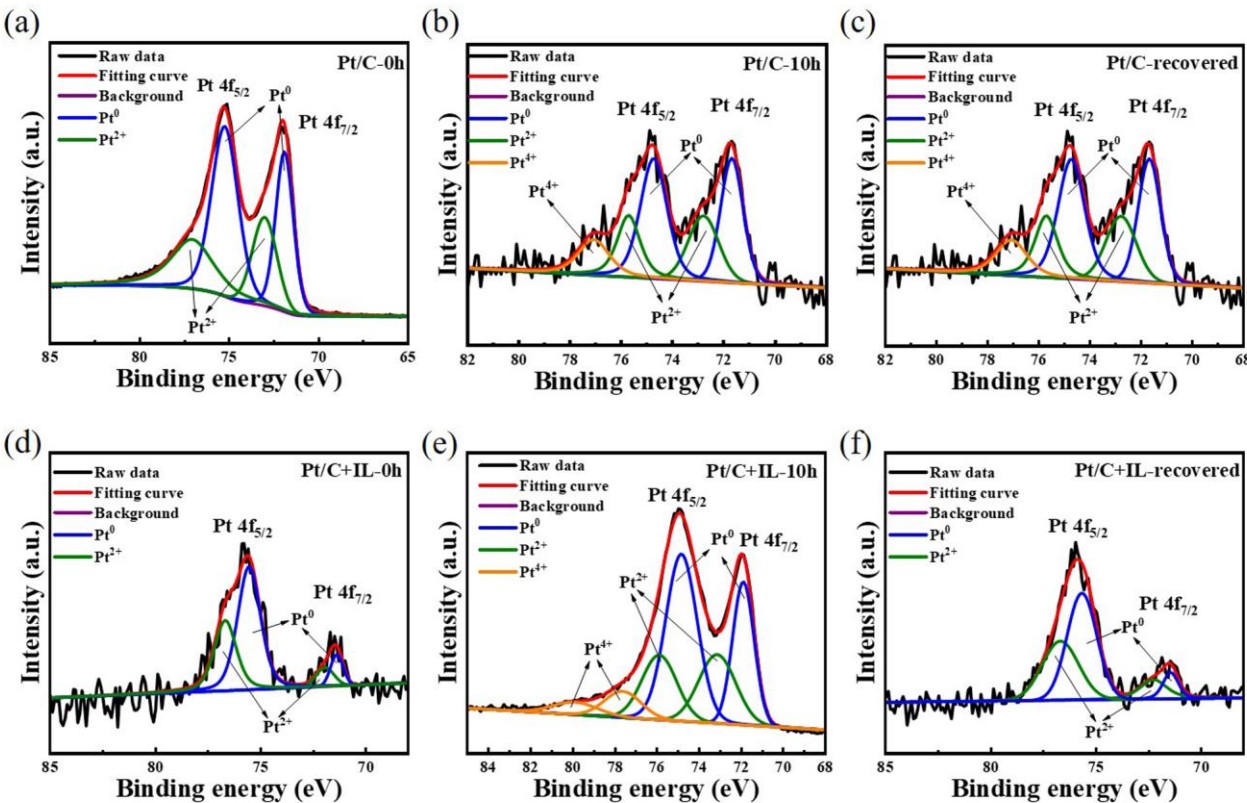

**Figure 6.** The Pt 4f XPS spectra for Pt/C before durability test (**a**), after durability test (**b**), and after CV recovery (**c**); for Pt/C + IL before durability test (**d**), after durability test (**e**), and after CV recovery (**f**).

**Table 1.** The relative peak areas of Pt and Pt + IL.

| Catalyst | Time | Relative Peak Area (%) | | |
|---|---|---|---|---|
| | | Pt Metal | Pt (II) | Pt (IV) |
| Pt/C | 0 h | 61.59 | 38.42 | - |
| | after 10 h | 55.84 | 32.52 | 11.64 |
| | recovered | 56.4 | 34.22 | 9.39 |
| Pt/C + IL | 0 h | 64.31 | 35.69 | - |
| | after 10 h | 57.42 | 34.38 | 8.21 |
| | recovered | 59.32 | 40.68 | - |

## 3. Experimental Part

### 3.1. Hydrophobic Treatment of Pt/C Catalyst

The hydrophobic treatment of the CL was carried out by dipping method, and the procedure was performed as in previous work [47]. At first, 0.512 g [BMIM][NTF$_2$] (98%) (from Sigma-Aldrich, St. Louis, MO, USA) was dispersed in 0.3 g IPA. Then, 1 g Pt/C catalyst (60TPM), wetted with a little DI water, was mixed with the above IL solution. The weight ration of IL and carbon support (IL/C) was 1.28. After that, the mixture was ultrasonically stirred for 20 min, evaporating the solvent slowly in the ambient atmosphere at 45 °C. Finally, the obtained powder was further dried overnight under high vacuum (-1 bar, room temperature) to obtain hydrophobic catalyst powder (Pt/C + IL).

### 3.2. Preparation of MEAs

The carbon electrodes were prepared from Vulcan XC-72 (Cabot, Boston, MA, USA), and the Pt electrodes were made by commercial Pt/C catalyst (60% Pt supported on

Vulcan XC-72). All samples were entrusted to WUT Energy Co., Ltd. (Wuhan, China) for processing. Firstly, the mixture consisting of Pt/C + IL (Pt/C), Nafion (5 wt%), isopropyl alcohol (IPA) and deionized water was put in a jar mill can and ultrasonically stirred for 1.5 h. The CLs were prepared by scraping the catalyst ink on the PTFE film and controlling the Pt loading at 0.4 $mg_{Pt}/cm^2$. Anode electrodes were prepared with Pt/C (TEC10V30E) dispersed in DI:IPA (70 wt% water) with an I/C of 0.9 (Nafion:C mass ratio) and loading at 0.1 $mg_{Pt}/cm^2$. Pt loadings on each individual electrode were verified by X-ray fluorescence spectroscopy (XRF) (Fischer XDV-SDD, Helmut Fischer, Sindelfingen. Baden-Wuerttemberg, Germany). Then, the CLs were dried in an oven at 80 °C for 30 min and cut into $5 \times 5$ $cm^2$ squares. They were transferred onto GORE film (12 μm) to make CCMs. The CCMs were sandwiched between two $5 \times 5$ $cm^2$ XGL gas diffusion layers (GDLs) at 15% compression. The MEAs were composed of the CCMs and clamping GDLs.

*3.3. Electrochemical Measurements*

**Condition procedure.** All cell measurements were conducted on an automated Greenlight Innovation fuel cell test station (Type G20, Greenlight Innovation, Burnaby, Canada). All the MEAs were activated before performance testing via a power holding step for 1.5 h at 80 °C, 150 $kPa_{abs}$, and fully humidified in hydrogen and air. When the I–V curve coincided twice, it was considered that the activation is sufficient. Then, the polarization curves were tested under 80 °C, 150 $kPa_{abs}$ and 100% RH with a stoichiometric flow of 1.5/2.0 ($H_2$/air).

**Electrochemical properties.** The CV curves were recorded with a scanning rate of 20 mV/s from 0.06 to 1.2 V under 30 °C and 100% RH. The Coulombic charge for $H_2$ adsorption was used to determine the electrochemical surface area (*ECSA*). According to the measured CV curve (hydrogen adsorption peak area), the *ECSA* can be calculated using Equation (6):

$$ECSA = \frac{S}{2.1 \times V \times M} \tag{6}$$

*V* is the scan rate (V/s), *ECSA* is the electrochemical active area ($m^2/g_{Pt}$), *S* is the integral area between the current and voltage in the hydrogen adsorption area (mA·V) and *M* is the Pt loading (mg).

**Pt oxide coverage test.** The test conditions and method for Pt oxide coverage were as referred to in earlier works [65]. The voltage was scanned from 0.85 V to almost 0.05 V and then scanned back to 1.2 V when recording the CV plots. The Pt oxide coverage at 0.85 V was determined by dividing the charge under the Pt oxide region ($Q_O$) by the charge under the H adsorption region ($Q_H$) [9]. The initial testing results were labeled Pt/C-0 h and Pt/C + IL-0 h, respectively.

**Durability procedure.** The single cells' operating conditions were at a constant potential of 0.85 V for 10 h at 80 °C, 150 $kPa_{abs}$ and 100% RH with a stoichiometric flow of 1.5/2.0 ($H_2$/air). After the durability test, the above polarization curve test, CV curve and Pt oxide coverage test were repeated again, marked as Pt/C-10 h and Pt/C + IL-10 h.

**Recovery procedure.** Finally, following the recovery procedure in reference [60], the CV technique from 0.06 to 1.2 V at 20 mV/s for 20 cycles was used to remove the oxide and recover the reversible loss. The electrochemical property tests were repeated, and the results were marked as Pt/C-recovered and Pt/C + IL-recovered.

*3.4. Characterization of Basic Physical Properties*

The contact angle was measured by a Contact Angle Meter (JC2000D4A, Powereach, Shanghai, China) to characterize the hydrophobicity of the CL. Measurement range of contact angle was 0~180°, and measurement accuracy was ±0.1°.

The X-ray photoelectron spectroscopy (XPS) test was carried out on the X-ray photoelectron spectrometer VG Multilab 2000 produced by Thermoelectron Corporation (Waltham, MA, USA). The excitation source was AlKα X-ray (1486.6 eV). The sample inclination was 65°, and the scanning step was 0.05 eV. The measurement proceeded

at $1 \times 10^{-6}$ Pa vacuum. All the binding energy was corrected by using the binding energy of C1s (284.6 eV).

## 4. Conclusions

In this study, [BMIM][NTF$_2$] was chosen to modify the surface of a Pt catalyst. Due to its hydrophobic and special ligand-like effect on Pt NPs, [BMIM][NTF$_2$] creates a hydrophobic, antioxidant micro-environment in the CL. In the single-cell test, the Pt/C + IL catalyst showed excellent electrochemical stability. The MEA performance of Pt/C + IL decreased by 45% at high constant potential (0.85 V) for 10 h, compared with the 60% of the untreated commercial Pt/C catalyst. The Pt/C + IL catalyst also showed excellent oxidation resistance at 0.85 V. After 10 h, the Pt oxide coverage of Pt/C + IL increased by 80.6%, while the commercial Pt/C catalyst increased by 263.2%. Thus, it can be seen that the improvement of high constant potential durability by [BMIM][NTF$_2$] is mainly due to its inhibition of Pt oxidation. The rapid increase in Pt oxide not only occupies the active sites, but also leads to Pt loss, which greatly reduces the performance of the catalyst. Furthermore, XPS results further confirmed that the total oxidation amount in the CL decreased after adding [BMIM][NTF$_2$]. After recovery by CV method, there was almost no Pt$^{4+}$ in Pt/C + IL, and there was still 9.39% Pt$^{4+}$ in the commercial Pt/C catalyst, which is mainly used as an irreversible stable oxidation substance to hinder mass transfer. Combined with the above advantages, the Pt/C + IL catalyst prepared in this work can effectively improve the catalyst stability via enhancing Pt antioxidant properties and, thus, improve the utilization rate of Pt.

**Author Contributions:** L.H. carried out all the experimental work and draft writing. F.Z., H.Z., J.T. and M.P. provided the writing, review and supervision. All authors have read and agreed to the published version of the manuscript.

**Funding:** This work was financially supported by National Natural Science Foundation of China (program no. 22109122 and 22209026).

**Data Availability Statement:** The data presented in this study are available in [this article].

**Conflicts of Interest:** The authors declare no conflict of interest.

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
