# Peer review of "Mechanistic Studies of Improving Pt Catalyst Stability at High Potential via Designing Hydrophobic Micro-Environment with Ionic Liquid in PEMFC"

_catalysts, doi:10.3390/catal13020374_

Round 1
Reviewer 1 Report
The manuscript entitled “Mechanistic studies of improving Pt catalyst stability at high potential via designing hydrophobic micro-environment with ionic liquid in PEMFC” used an ionic liquid [BMIM][NTF2] to modify the Pt catalyst (Pt/C+IL) to create a hydrophobic antioxidant micro-environment in the catalyst layer (CL). And this work aims to explore the influence mechanism of the hydrophobic micro-environment created by IL on Pt oxidation and the relation between Pt oxidation and durability at high constant potential. In general, the research content of this paper is logically rigorous, and the data is sufficient to support the conclusions in the paper. At the same time, the durability of catalytic layer is indeed a core research focus of PEMFC heavy-duty vehicle, and the research points of this article is useful and instructive. Thus I recommend that the article can be published after a minor revision.
1. The introduction to the modification of the catalytic layer is too refined, and the overview of the application of ionic liquid in PEMFC should be increased.
2. Both Figure 3 and Figure 5 are about CV curves, but the test conditions should be different, which needs to be written clearly in the experimental section. In particular, the absence of the oxidation curve in the high voltage region in Figure 3 should also be explained as necessary.
3. Abstract should be further improved. The abstract now feels like a deliberate attempt to obscure the conclusion due to word limits. This article deserves a concise abstract with salient points
4. The conclusion section should not be a list of results, more mechanism analysis should be added.
Reviewer 2 Report
1. Page 3, line 108, 2.2 Preparation of MEAs. The preparation method should be explained in more details, some missing information is necessary.
2. Page 4, Figure 1, line 151. The explanation in the text is incomplete; Authors should explain why the higher contact angle after treatment increases the hydrophobicity of Pt/C catalyst layer.
3. The conclusion part is written as points. It should be more representative of the results and be rewritten as a conclusion following the general format of this journal.
4. I found some grammatical mistakes, some corrections are needed.
